# Targeting Epigenetic Changes Mediated by Members of the SMYD Family of Lysine Methyltransferases

**DOI:** 10.3390/molecules28042000

**Published:** 2023-02-20

**Authors:** Alyssa Padilla, John F. Manganaro, Lydia Huesgen, Deborah A. Roess, Mark A. Brown, Debbie C. Crans

**Affiliations:** 1Department of Biomedical Sciences, Colorado State University, Fort Collins, CO 80523-1617, USA; 2Department of Chemistry, Colorado State University, Fort Collins, CO 80523-1872, USA; 3Cell and Molecular Biology Program, Colorado State University, Fort Collins, CO 80523-1005, USA; 4Department of Clinical Sciences, Colorado State University, Fort Collins, CO 80523-1678, USA; 5Graduate Degree Program in Ecology, Department of Ethnic Studies, Global Health and Health Disparities, Colorado School of Public Health, Colorado State University, Fort Collins, CO 80523-1612, USA

**Keywords:** SMYD proteins, lysine methyltransferases, epigenetic drugs, cancers, SMYD SET, MYND, repressed transcription of SMYD genes, leukemia, breast cancer, cardiac tissue

## Abstract

A comprehensive understanding of the mechanisms involved in epigenetic changes in gene expression is essential to the clinical management of diseases linked to the SMYD family of lysine methyltransferases. The five known SMYD enzymes catalyze the transfer of donor methyl groups from S-adenosylmethionine (SAM) to specific lysines on histones and non-histone substrates. SMYDs family members have distinct tissue distributions and tissue-specific functions, including regulation of development, cell differentiation, and embryogenesis. Diseases associated with SMYDs include the repressed transcription of SMYD1 genes needed for the formation of ion channels in the heart leading to heart failure, SMYD2 overexpression in esophageal squamous cell carcinoma (ESCC) or p53-related cancers, and poor prognosis associated with SMYD3 overexpression in more than 14 types of cancer including breast cancer, colon cancer, prostate cancer, lung cancer, and pancreatic cancer. Given the importance of epigenetics in various pathologies, the development of epigenetic inhibitors has attracted considerable attention from the pharmaceutical industry. The pharmacologic development of the inhibitors involves the identification of molecules regulating both functional SMYD SET (Suppressor of variegation, Enhancer of Zeste, Trithorax) and MYND (Myeloid-Nervy-DEAF1) domains, a process facilitated by available X-ray structures for SMYD1, SMYD2, and SMYD3. Important leads for potential pharmaceutical agents have been reported for SMYD2 and SMYD3 enzymes, and six epigenetic inhibitors have been developed for drugs used to treat myelodysplastic syndrome (Vidaza, Dacogen), cutaneous T-cell lymphoma (Zoinza, Isrodax), and peripheral T-cell lymphoma (Beleodag, Epidaza). The recently demonstrated reversal of SMYD histone methylation suggests that reversing the epigenetic effects of SMYDs in cancerous tissues may be a desirable target for pharmacological development.

## 1. Introduction

SMYDs are a family of five unique lysine methyltransferases (PKMTs) that catalyze the transfer of donor methyl groups from S-adenosylmethionine (SAM) to specific lysines on histones and non-histone substrates [1,2,3,4,5,6]. The resulting changes in chromatin structure and altered transcriptional regulation can lead to diseases or be associated with poorer outcomes for specific cancers [1,2,3,4,5,6,7,8,9]. SMYD proteins contain both functional SET (suppressor of variegation, enhancer of zeste, Trithorax) and MYND (Myeloid-Nervy-DEAF1) domains (Figure 1) [1,2,6,10,11]. As shown in Figure 1 and reactions (1) and (2) [9], lysine residues can be alkylated up to three times depending on the enzyme catalyst. As an example, SMYD5 has been reported recently to monoalkylate histone H3 on the lysine residue 36 (H3K36me1) in mammalian cells in vitro [4], and also to trialkylate H3 on lysine residue 36 (H3K36me3) at promotors regulating gene expression [5]. More generally, SMYD lysine methyltransferases target specific residues on histones to either silence or enhance the expression of associated genes through the post-translational methylation of the histone tails. Of pharmacologic interest, SMYD-mediated changes in chromatin structure and altered transcriptional regulation may be treatable in, for example, drug-resistant cancers [12] using newly developed therapies to reverse epigenetic changes in chromatin structure [7,8,13,14,15]. 

SMYD-mediated enzyme reactions require the activation of a methyl (Me, CH_3_)-group and a specific transfer from SAM to a peptide containing a lysine, as shown in reaction (1) in Figure 1. SMYD enzymes bind protein substrates, which direct the Me-group transfer, to the correct lysine group on the histone. As shown in the Figure 1 reaction (2), SMYD enzymes can transfer one or more Me groups to a particular lysine [9] catalyzing mono, di, and tri-methylation [9,12] which, in turn, provides much more regulatory control. Importantly, reaction (2) shown in Figure 1 is reversible; proteins can be demethylated by protein lysine demethylates (PKDM) [9], which are a novel class of enzymes [9]. The existence of enzymes capable of reversing SMYD-mediated reactions demonstrates a critical difference between epigenetic post-translational modifications and more permanent mutational changes in protein sequences [11].

The first member of the SMYD family of proteins, SMYD 1, was discovered in 1995 in a reverse reading frame of CD8b [2]. Since then, four other SMYD family members have been identified [6,16,17,18,19,20]. The domain structures for the five known SMYDs are shown in Figure 1 [6,16,17,18,19]. Each member of the SMYD family contains a functional SET (suppressor of variegation, enhancer of zeste, Trithorax) domain which, as shown in Figure 1, consists of two noncontinuous elements, N-SET and C-SET. The N-SET and C-SET domains are highly conserved and separated by the MYND (Myeloid-Nervy-DEAF1) domain which contains a zinc-finger motif serving as an intermediary in protein–protein interactions through proline-rich regions [1,2]. Genes for SMYD1, SMYD2, SMYD3, and SMYD5 range from 400–500 residues, while the gene for SMYD4 is approximately twice that size, with SET and MYND domains positioned after a 226 residue tetratricopeptide (TPR) domain [6]. SMYDs also contain a post-SET domain that is rich in cysteines [1,21,22,23]. 

SET domain-mediated methylation events drive the epigenetic regulation of cellular processes involved in embryonic development and homeostatic function [1,24]. As examples, SET-mediated lysine methylation plays a crucial role in X chromosome inactivation, DNA methylation, chromatin accessibility, and the regulation of transcription [1]. SMYD lysine methyltransferases mediate post-translational methylation of histone tails to either silence or enhance the expression of associated genes [1,25,26]. Transcriptional activation is most often associated with lysine methylation at H3K4, H3K79, or H3K36 [1,26,27,28]. In contrast to their roles in the normal regulation of biological processes, the aberrant expression of SMYD proteins results in pathophysiological events, including the epigenetic induction of oncogenesis [2,12,29].

Structural information is available for SMYD1, SMYD2, and SMYD3 proteins which share clear similarities [29] and, in general, there is a high degree of homology in their gene sequences (Figure 1) [30,31,32]. Like SMYD1 and SMYD2 [9], SMYD3 proteins consist of six sub-domains, N-SET, MYND, I-SET, core-SET (also referred to s C-SET or (S)ET in the literature), post-SET, and C-terminus domains (also referred to as C-lobe or CTD). The SMYD3 N-SET domain is highly conserved with other SET proteins found in humans, plants, yeast, and viruses [24,29,32]. The SYMD3 C-terminus domain is the least conserved among SMYD proteins, with no significant similarity to other known protein domains. Interestingly, a RUBICO_LSMT domain referred to as I-SET has been identified in the SMYD3 protein. This I-SET domain is composed of four beta sheets linked by a hairpin, between beta-1 and beta-2, and a loop between beta-2 and beta-3 [29,32]. SMYD3’s MYND domain binds Zn^2+^ in a zinc-finger domain that links its Rubisco-LSMT, SET-N, and SET-C domains. The protein binding pocket is near the SET-C and the post-C domains [29,32].

Although each SMYD methylates histones, the tissue distribution and function of the individual SMYD proteins varies [16,17,18]. Some SMYDs are found only in cancerous cells, representing potential targets for pharmacologic agents, while others have been identified in both healthy and cancerous tissues. Because SMYD proteins play a role in gene expression regulation in host immunity against pathogen infection, hematopoiesis, and several cancers, there are obvious and less obvious relationships between SMYDs and immune system function [3]. SMYD overexpression can be linked to several diseases. Overexpression of SMYD1 represses the transcription of genes, leading to the formation of ion channels in the heart, a process which causes heart failure [33]. SMYD2 is overexpressed in esophageal squamous cell carcinoma (ESCC) and triple-negative breast cancers, which usually contain TP53 missense mutations and high levels of p53. A knockdown of SMYD2 inhibits tumor cell proliferation [33,34]. SMYD3 is overexpressed in cancers, including breast cancer, colon cancer, prostate cancer, lung cancer, and pancreatic cancer [35,36,37,38,39,40], where it correlates with poor prognosis. 

Although no SMYD inhibitors have advanced to clinical trials [41], there are, depending on the SMYD enzyme, several clinically important drugs, including doxorubicin and cisplatin, where the development of drug resistance can be countered by SMYD inhibitors [12]. Some novel candidates including AZ505, BCI-121, and EPZ031686 have been reported to effectively inhibit SMYD methylation reactions [42,43,44]. The potential for the therapeutic targeting of SMYD function has been investigated by several industrial companies, leading to thousands of compounds being tested and reports made of key target compounds. Early leads include AZ50552 [45] for SMYD2 and BCI-121 [42] and EPZ031686 [46]) for SMYD3, each with nanomolar IC_50_ values, as is discussed in more detail below. Six epigenetic inhibitors have been developed for drugs [47] and are currently in use in the clinics where they are being used to treat myelodysplastic syndrome (Vidaza [48], Dacogen [49]), cutaneous T-cell lymphoma (Zoinza [50], Isrodax [51]) and peripheral T-cell lymphoma (Beleodag [52], Epidaza [53]).

In this review, a summary of each SMYD member is presented, together with experimental findings that suggest possible therapeutic targets for application. 

## 2. Approaches to the Development of Pharmacologic Agents Modulating SMYD Function 

The development of a therapeutic agent involves the identification of an inhibitor for the epigenetic function of the specific SMYD protein under investigation [45,54,55]. If the target protein is an enzyme which catalyzes a reaction, the first step is the identification of a suitable substrate for use in assays selected to screen potential drug candidates [45,47]. This requires knowledge of the specific cellular mechanisms utilized by individual members of the SMYD family [15] and of the combined tools of the biologist and medicinal chemist. For SMYDs, this involves the identification of potential peptides containing lysine residues with sequences similar to those in naturally occurring histones or non-histone proteins targeted by SMYDs and syntheses on a scale suitable to carry out the needed studies. Considering that histones may be different depending on the cell types investigated, determining mechanisms involved in the inhibition of the enzymatic activity of the SMYD proteins is not a trivial goal and underscores the importance of life scientists collaborating with chemists to develop potential targets for the treatment of the identified lethal malignancies. 

### 2.1. Inhibitor Development Based on Structure–Activity Relationships (SAR)

Structure–activity relationships (SAR) are used to identify inhibitors for enzymes and to carry out a detailed searches for potential therapeutic agents. Often, an initial screening identifies a lead which is then explored further using SAR methods [43,56,57]. The objective is to systematically probe different structural parts of the lead and thus explore the structurally related compounds within a particular chemical space to optimize the biological effects of the lead compound. The medicinal chemist is guided by the approaches described by Lipinski’s rule of five [58] when modifying the structure of the lead compound. It is important to recognize that this fragment-based method can affect the quality of potential hits [59]. The early identification of false hits saves resources pursuing compounds involved in multiple metabolic pathways and which are unlikely to lead to potential drugs. With the development of combinatorial libraries, as well as computational methods, it is possible to evaluate larger parts of chemical space for new leads. This method contrast strategies in which one inhibitor is investigated at a time. These methods have been used in industrial settings, as well as in medicinal chemistry programs, to examine thousands of potential compounds for the development of leads.

Once leads have been identified in in vitro, ex vivo, and in vivo studies, suitable animal model systems follow. Such studies require pharmacological information on the properties of the compound such as cellular adsorption, distribution, metabolism, and excretion (ADME), as well as a detailed study of pharmacokinetic and pharmacodynamic processing in suitable animal model systems. Together with toxicity information, these data guide which compounds will be selected for further investigation and, ultimately, human studies and clinical trials. 

### 2.2. In Silico Methods for Inhibitor Identification to Structurally Known Proteins

Biologists and computational scientists have developed an alternative in silico approach, based on computational analysis, to screen large libraries of compounds. In this approach, screening could be performed without bench testing thousands of compounds and without the need for initial synthetic efforts by medicinal chemists. Using an in silico method based on the hit-to-lead methodology, the binding affinity can be determined from a very large, known library of compounds, a major advantage of using this method. An example of a library developed for this purpose is the library of 137,990 molecules of the Small Molecule Drug Discovery Suite (Schrodinger, Inc., New York, NY, USA) from the free ZINC15 database. This library and others like it include entirely accessible and “purchasable” molecules. After the reported in vitro screens have been carried out and the structures are identified in the ZINC15 database [60], compounds can then be docked into a protein X-ray structure [6]. A suitable X-ray structure of an SMYD protein is, for example, the SMYD3 PDB structure, with identification code 5EX3, from the Protein Data Bank (PDB) [61]. One can upload appropriate small molecule structures to test their affinity to the protein and identify the substrate binding site. Being mindful of where compounds bind to the target protein is important because two different approaches can be taken. One approach is to identify compounds interacting with the SAM binding pocket on SMYDs, such as PRMT5 and PRMT7, which function as selective competitive inhibitors for SAMs and other methyltransferases [62]. A second approach is to focus on the compounds associated with the substrate binding site. After docking each molecule screened into SMYD3′s protein-substrate binding pocket, the top hits were identified as those with the most negative binding energy and were confirmed as a lead using the ZINC15 molecular similarity search engine. 

An example of the application of the in silico/in vitro hit-to-lead enzyme inhibition platform was employed to identify a small molecular inhibitor for SMYD3 lysine methyltransferase activity [43]. Compounds were evaluated by searching libraries of small molecules for compounds with high binding affinity for SMYD3. Promising lead compounds were then used in binding assays in vitro with breast cancer cell lines [43]. This analysis identified several inhibitors for SMYD3 [43]. From the top 10 hits, the 50 most similar compounds were scored using Schrodinger software, a process examining a total of 500 compounds. Five small molecule inhibitors, designated inhibitor-1 through inhibitor-5, were identified as having the potential to disrupt uncontrolled oncogenic breast cell proliferation without affecting normal cell function and were tested in vitro using an assay of SMYD3 methyltransferase activity. One small molecule inhibitor, designated inhibitor-4, disrupted uncontrolled oncogenic breast cell proliferation without affecting normal cell function in a human epithelial breast cancer cell line. Inhibitor-4 was further advanced to other human cell line experiments [43] using lung (A549) and colorectal (DLD-1) cells [32]. The identification of this inhibitor enables further studies of structure–activity relationships (SAR), as described above, for the further development of improved inhibitors. 

## 3. SMYD-Proteins, Structure, Function, and Medicinal Potential

Experimental approaches to the development of SMYD inhibitors via studies of structure–activity relationships or in silica methods depend on a detailed understanding of SMYD structure and function and epigenetic effects [11]. The SMYD family of lysine methyltransferases contains a functional SET domain which, as shown in Figure 1 consists of two noncontinuous elements from the N- and C-terminal ends of the SET sequence. In SMYD1, SMYD2, SMYD3, and SMYD5, the genes range from 400–500 residues, whereas SMYD4 is twice that length, as shown in Figure 1. Each of these SMYD proteins has been found in many tissues where SET domain-mediated methylation functions in disease drive the regulation of cellular processes essential to development or homeostatic function [1,24]. X-ray structures of SMYD1, SMYD2, and SMYD3 are shown to be bound to a cofactor analog Sinefungin (SFG) or S-adenosyl-L-homocysteine (SAH) in Figure 2. N-SET and C-SET domains are highly conserved and are separated by an MYND domain [1,2]. The structural motif begins with the S domain (light green), which consists of beta sheets and random coil domains. This is followed by the MYND (blue) domain, with a random coil region leading into an alpha helix that binds two Zn^2+^ ions (green). The post-SET domain (cyan) comprises mainly alpha helixes. The CTD C-terminus (red) consists mainly of alpha helixes and a small amount of random coil. SMYD1 is shown in Figure 2 in an open conformation. SMYD2 is shown in an intermediate conformation bound to both SFG and SAH. SMYD3 is shown in a closed conformation bound to SFG [29]. Finally, a composite image illustrates the movement of the C-Lobe from the open through the intermediate to the closed conformation and shows how the C-Lobe (C-terminal) changes through these movements.

The SMYD3 enzyme is organized sequentially from CTD, used for protein–protein interactions, to SET, which is responsible for lysine methylation, to SET-I, a MYND containing a zinc-binding motif for protein–protein interactions which prefers binding to proteins with proline-rich regions, to a linker and finally to a portion of post-SET. This organization differs from that of SMYD1 [63]. While the crystallization and characterization of SMYD1 has been reported in an open conformation, the structures of SMYD2 and SMYD3 have only been reported in intermediate and closed conformations. The MYND, SET, and SET-I domains are linked and separate from post-SET and CTD domains, depending on the distance from the N lobe to the CTD terminal domain. Interestingly, SMYD1, in the open conformation, showed a different conformation when compared to the more closed conformations of SMYD2 and SMYD3, suggesting that substrate selectivity may be affected by enzyme conformation. 

In Table 1, we summarize studies that are helpful in understanding the function and selectively of the SMYD proteins, their structures, SMYD inhibitors that have been developed, and targeted epigenetically induced diseases. X-ray crystallographic studies have been carried out for SMYD1 [29,63], SMYD2 [29,45,64,65,66,67], and SMYD3 [29,31,68,69]. At this time, no structural studies have been reported for SMYD4, but some work is underway for SMYD5 [70].

### 3.1. SMYD1

#### 3.1.1. Structure and Function of SMYD1

SMYD1, discovered in 1995 [2], can be categorized as a muscle-specific histone methyltransferase linked to the epigenetic regulation of cardiomyocyte differentiation and cardiac energy metabolism [16,17,18]. Studies of SMYD1 have focused primarily on its role in the proper development of the adult mammalian heart and associated muscle tissue. More recently, SMYD1 has been shown to function in sarcomere organization in cranial, facial, and extraocular muscles [89]. SMYD1 is unique to the SMYD family members due to the high expression of multiple isoforms in cardiac and skeletal muscle tissue and its expression in CD8^+^ T lymphocytes [18,90]. The regulation of metabolic energetics of cardiac cells is coupled with a central regulator of mitochondria, peroxisome proliferator-activated receptor gamma coactivator 1-alpha (PGC-1α), to mediate control of transcription [16,18]. The catalytic activity of this protein may also be controlled by a distinctive C-terminal domain that assumes two different structural configurations depending on environmental conditions. Additionally, it may assist in the stabilization of the SET domain [1,16,17]. SMYD1 facilitates the methylation of H3K4, resulting in gene-specific initiation of transcription; the interaction of the MYND domain with histone deacetylase (HDAC) 1, 2, or 3 is responsible for further regulation of the associated gene expression [18,91]. An interface with the muscle-specific transcription factor, skNAC, via the SMYD1 MYND domain further mediates the regulation of downstream transcriptional targets [18]. 

Recent studies have demonstrated that SMYD1 is necessary for proper myogenesis, muscle contraction, and myofibril arrangement (Figure 3) [16,17,91]. When this protein is not properly expressed, ventricular maturation is disrupted. This appears to be related to reduced methylation of the cardiac endoplasmic reticulum stress metabolic sensor, TRB3, by SMYD1 [16,91]. In association with skeletal muscle development during sarcomerogenesis, SMYD1 co-localizes with emergent myosin structures along the M-line of developed sarcomeres [18,89,90,91]. 

In mouse models, an SMYD1 knockout from early precursors of muscle cells using Myf5^cre^ altered cell differentiation but did not affect cell proliferation [18,92]. A knockout of SMYD1 and skNAC using directed siRNA in C2C12 myocytes (an immortalized mouse myoblast cell line) provided evidence that the integration of this complex was involved in the increased di- and tri-methylation of H3K4 due to a significant decrease in methyl transfer at calpain 1 and SRF promoters. The mono-methylation of this site was not affected (Figure 4) [93]. In addition, the deletion of SMYD1 decreased histone deacetylase activity in knockout cells, suggesting that this protein is an activator of histone deacetylase activity. 

Other studies examining muscle-specific chromatin regulators have indicated that SMYD1 is linked to the development of hypertrophic cardiomyocytes, leading to heart failure in mice [92]. The elimination of SMYD1 caused the overgrowth of cells and organ remodeling, suggesting that SMYD1 is necessary for mediating the growth restriction of the heart [92]. Furthermore, SMYD1 expression was decreased in the myocytes of sepsis-induced cardiomyopathy (SIC) patients, LPS-induced SIC rats, and LPS-induced H9c2 cardiac cells (Figure 4) [94]. These data suggest that the skNAC-SMYD1 complex is involved in transcriptional regulation both through the control of histone methylation and histone (de)acetylation.

#### 3.1.2. Therapeutic Applications, Potential Drug Targeting, and the Use of SMYD1 as a Prognostic Indicator

Although no drugs or inhibitors have been developed that specifically target SMYD1, the modulation of SMYD1 function can be demonstrated. Anthracycline doxorubicin (DOX), widely used to treat various cancers, is associated with the cumulative and dose-dependent potential for cardiotoxicity and congestive heart failure [71]. The exposure of cultured cardiomyocyte precursor cells to DOX causes apoptotic cell death associated with cellular oxidative stress. In addition, the regulation of proteins involved in epigenetic processes takes place and changes in global histone acetylations occur. A study with human pluripotent stem cell-derived ventricular cardiomyocytes in 3D-engineered cardiac tissue demonstrated overexpression of SMYD1 in cardiac microtissue and tissue strips [95]. SMYD1 was found to have a role in cardiac gene expression, contractility, Ca^2+^ handling, electrophysiological functions, and in vitro maturation.

The structure, function and methylation targets of the SMYD family of proteins play prominent roles in cardiac and skeletal muscle physiology and pathology [6]. Specifically, the SMYD1 and the skNAC proteins are transcription factors in hematopoiesis and cardiac/skeletal muscle (93, 96). A comparative analysis of genes deregulated by an SMYD1 or skNAC knockdown in differentiating C2C12 myoblasts led to the identification of the transcript characteristic of neurodegenerative diseases, including Alzheimer’s, Parkinson’s and Huntington’s Diseases [96]. Simplified, Alzheimer’s destroys memory, while Parkinson’s and Huntington’s affect movement. However, all three are caused by the death of neurons and other cells within the brain. Based on meta-analyses and direct experimentation, SMYD1 and skNAC expression within the cortical striata of human brains, mouse brains and transgenic mouse models of these diseases was reported [96]. These features were observed in mouse myoblasts which were induced to differentiate into neurons. For example, the defining features of the pathology of Alzheimer’s, including the brain-specific, axon-enriched microtubule-associated protein, tau, was found to be deregulated upon loss of SMYD1 [96].

These results were consistent with the notion that epigenetic priming requires the triggering of signals such as those existing in a 3D environment. In another approach, all five SMYD proteins were tested for effects on the histone methyltransferase family of enzymes, including on histone lysine methyltransferases (HKMTs) and histone/protein arginine methyltransferases (PRMTs) [97]. Only the SMYD1 gene had prognostic properties and also seemed to play a role in the pathogenesis of prostate cancer [97] in addition to its role in cardiac function [2,16,17,18].

### 3.2. SMYD2

#### 3.2.1. Structure and Function of SMYD2

SMYD2 has been identified in normal tissues as well as in tumor cells [10,78]. In mice models, SMYD2 is expressed in a range of tissues such as the liver, kidneys, thymus, hypothalamus, vomeronasal organ, and ovaries [73,78]. Methyltransferase activity in these tissues occurs via the highly selective methylation of both histone and non-histone proteins such as retinoblastoma protein (RB) and tumor suppressors (p53) [73,74]. SMYD2 is involved in a wide range of cancers, causing cell proliferation through mechanisms that include various functionally independent cellular processes and nonhistone lysine substrates [12]. Nevertheless, a complete knockdown of SMYD2 may not be desirable since SMYD2 inhibition can have dramatic and perhaps lethal side effects [98]. A knockdown of SMYD2 has demonstrated a tendency to reverse HIV latency. Thus, designing a therapeutic drug that selectively inhibits SMYD2 function in cancer cells may require the identification of targets in only a subset of SMYD2-expressing cells. As an example, the binding of a small molecule to the proline-rich sequences of MYND may be possible in cells where the tumor suppressor EBP41L3 links to SMYD2, as is the case in meningiomas, a brain cancer, and lung cancer [10,23,99,100]. The investigation of the structural differences between SMYD proteins will be necessary to establish specificity and assist drug design. The mechanisms for methylation of nonhistone lysine substrates are summarized in Figure 5 [9]. 

SMYD2 is necessary during the development of the heart and brain and functions via H3K4-specific methyltransferase activity [76]. The activation of Wnt signaling is initiated by the methylation of β-catenin by SMYD2, which then commits pluripotent stem cells to their designated mesendoderm fate [73]. The methylation by SMYD2 functions in the skeletal myocyte cytoplasm through the targeting of protein chaperone Hsp90 [73,74,76,99,101]. The methylation of the cytoplasmic protein chaperone induces the formation of SMYD2, Hsp90, and the sarcomeric protein titin complex in muscle, indicating a role for SMYD2 in the maintenance of the skeletal muscle integrity [73,101]. The SMYD2-Hsp90 methylated complex preserves the stability of the skeletal muscle by binding to the N2A-domain of the titin to protect myocytic placement and the sarcomeric I-band area [76,101]. 

In addition to its functions in muscle tissues, SMYD2 contributes to the regulation of hematopoiesis [75,102]. The production of hematopoietic stem cells, derived from progenitors of hemogenic endothelial cells originating in bone marrow, depends on SMYD2-mediated methylation [102]. The loss of SMYD2 function leads to a decrease in hematopoietic stem cells due to apoptosis [102]. The disruption of hematopoietic stem cell transcription and altered Wnt signaling, by way of β-catenin, may interfere with differentiation and contribute to the development of leukemias, implicating SMYD2 as an oncogene [75]. In addition, when SMYD2 is eliminated in the hematopoietic stem cells of fully grown mice, there is an increase in apoptosis and transcriptional defects in stem cells, further demonstrating the importance of this enzyme throughout development and in adult organisms [73,76,102].

It is proposed that SMYD2 suppressed p53-dependent apoptosis by methylating the p53 transcription factor at lysine 370 in cardiomyocytes, as shown in Figure 6 [103]. This results in the separation of p53 from target gene promoters including p21, a cyclin-dependent protein kinase that modifies the regulation of cell cycles [103]. Because methylation of p53 serves as a protective mechanism to inhibit apoptotic cell death, targeting the methylation of p53 may prove to be a therapeutic strategy to limit cell loss in diseased states such as heart failure [103]. 

Recently, SMYD2 and SMYD3, together with other SET proteins, have been shown to function in the development of vascular disease. The methylation of H3K4 and H3K36 was dependent on SET and associated with vascular calcification [104,105]. To evaluate the pathophysiology of vascular calcification, the development of pharmacologic agents that target and reverse the methylation of lysine groups on histones to restore homeostasis has been proposed. 

Based on high expression levels, the function of SMYD2 may contribute to the development and progression of leukemias including CML, ALL, B-ALL, MLLr-B-ALL, AML, and T-ALL [75]. Acute lymphoblastic leukemia (ALL) is associated with genetic aberrations, mutations, and chromosomal translocations during lymphocyte development (Figure 7) [19]. Overexpression of the gene for SMYD2 is linked to pathogenesis in adolescent B cell ALL (B-ALL) related to the MLL-AF9 oncogene and in triple-negative breast cancer with a poor patient prognosis [19,75]. SMYD2 mRNA expression in Leukemic bone marrow samples is abnormally high when compared with non-neoplastc samples, Figure 7 [19]. An SMYD2 knockout in MLL-AF9-induced leukemias resulted in dormancy of primary leukemia cells and decreased progression of the malignancy while not impacting hematopoiesis. These results have increased enthusiasm for SMYD2 as a target for therapeutic treatments in these diseases [19,105]. 

#### 3.2.2. Therapeutic Applications, Potential Drug Targeting, and Prognostics SMYD2

Two distinct chemical series of small molecules, a benzoxazinone series AZ50552 [45] and A-89361 [54] and a pyrrolidine series LLY-50762 [55]), have been reported to inhibit the methyl transfer reaction which is catalyzed by SMYD2 (Figure 2 and Figure 3). Despite differences in chemical structure, members of both the benzoxazinone and pyrrolidine series bind to SMYD2 similarly, associating with the surface of the lysine channel, the two hydrophobic pockets of the SAM binding site, and the protein binding site. The lead compounds are AZ50552 (also referred to as AZ505) [45,106] and A-89361 (also referred to as A-893) [54] for the benzoxazinone and LLY-50762 (also referred to as LLY-507) [55] for the pyrrolidine series. AZ50552 was selected from a high-throughput screen of 1200 compounds that generated 25 compounds with IC_50_ values less than 40 microM. AZ50552 (the official IUPAC name is [N-cyclohexyl-3-(3,4-dichlorophe-nethylamino)-N-(2-(2-(5-hydroxy-3-oxo-3,4-dihydro-2H-benzo[b][1,4]oxazin-8-yl)ethylamino)ethyl)propenamide) was initially reported to have an IC_50_ of 120 nM. In crystal structure analyses, AZ505 was shown to bind to the lysine access channel. ITC analysis indicated that inhibitor binding was primarily driven by hydrophobic interactions alone, providing a low KD ~500 nM. The potency of AZ505 was due to a complete blockage of the core region of SMYD2’s active sites, which prevented SMYD2 from binding to the target lysine. Further optimization of AZ505 showed that it worked to inhibit SMYD activity in several cancer cell lines [106]. Although these reports document the identification of lead inhibitors, there are ongoing efforts to identify new lead compounds that build on the earlier studies and to introduce new strategies for drug development [106]. 

The estrogen receptor (ER) is involved in signaling processes and has been implicated in diseases such as cancers [107]. The crystal structure of SMYD2, associated with a target lysine (Lys266)-containing ERα peptide, provides molecular information relevant to drug design. The structure shows that the ERα peptide is in a U-shaped conformation when its lysine binds to SMYD2 Figure 8. The structure shows intrapeptide contacts, demonstrating complementarity between the substrate and the active site of SMYD2, and allows comparison with the SMYD2–p53 structure, with insights into the diverse nature of SMYD2’s substrate recognition [107]. The broad specificity of SMYD2 appears to involve multiple molecular mechanisms. Surprisingly, a polyethylene glycol (PEG) binding site is identified in the CTD domain of SMYD2, extending potential interactions of this protein, and may impact substrate specificity and target binding diversity.

Two clinical roles for SMYD2, the reversal of drug resistance and its use as a prognostic indicator for disease progression, have been identified. Agents targeting SMYD2 may reverse cisplatin or oxaliplatin drug resistance. SMYD2 is implicated in cisplatin resistance through its regulation of the p53 pathway [108]. As a therapeutic target in non-small cell lung cancer (NSCLC), reducing SMYD2 activity via specific inhibitors appears to enhance the cell sensitivity to cisplatin but not to paclitaxel, vinorelbine, or vincristine sulfate [108]. This may be due to overexpression of SMYD2 and its substrates in NSCLC-resistant cells, where either the inhibition of SMYD2 or knockdown by specific siRNA reverses cell resistance to cisplatin treatment. The efficacy of treating colon cancer with a oxaliplatin, is similarly limited by the development of drug resistance [109]. The knockdown of SMYD2 increased sensitivity to oxaliplatin in vitro and in vivo, while SMYD2 overexpression promoted oxaliplatin resistance in vitro. SMYD2 upregulated MDR1/P-glycoprotein expression, depending on the MEK/ERK/AP-1 signaling pathway. In the treatment of gliomas, SMYD1-mediated degradation of SMYD2 played an important role in reversing resistance to chemotherapeutic agents. A tumor tissue microarray, using samples from 441 patients with glioma, was used to measure the presence of SMYD2 [41]. Interestingly, cisplatin treatment of AZ505-pretreated glioma cells caused a significant decrease in SMYD2 expression when compared to glioma cells that were not pretreated. More recently, SMYD2 was reported as a novel molecular target in metastatic castration-resistant prostate cancer. An ingredient in extra-virgin olive oil, S-(-)-oleocanthal, emerged as a specific SMYD2 lead inhibitor. This compound had high in vivo potency and a strong safety profile and has been proposed as a novel nutraceutical to treat metastatic castration-resistant prostate cancer [110].

SMYD2 has been demonstrated for several cancers as a prognostic indicator for disease severity and progression. The overexpression of SMYD2 has been linked to gastric cancer, where it inhibits the transactivation of p53, leading to larger and more invasive tumors [79]. This suggests that the continued proliferation of gastric tumor cells is connected to SMYD2 function and, as is the case for some leukemias, highlights the utility of SMYD2 as a prognostic indicator for disease progression [79]. SMYD2 was reported to be a prognostic indicator of clear cell renal cell carcinoma (ccRCC) progression and as also playing a role in tumorigenesis and multi-drug resistance [97]. The effects of SMYD2 and SMYD2-mediated miRNAs on renal cancer cell proliferation, migration, clonogenicity, and tumorigenicity were determined via cell-function assays and murine xenograft experiments. The effects of five antineoplastic drugs (cisplatin, DOX, fluorouracil, docetaxel, and sunitinib) were measured in AZ505-treated and control cells to verify the presence of multiple-drug resistance. However, as a caveat, a three-dimensional distribution of samples according to expression levels of SMYD2 could predict in vitro responsiveness to doxorubicin in women with breast cancer [111]. However, it could not predict the responsiveness to doxorubicin treatment in short-term cultures of dog mammary gland tumor slices [112].

### 3.3. SMYD3

#### 3.3.1. Structure and Function of SMYD3

The methyltransferase activity of the SET domain in the SMYD family was first demonstrated using SMYD3. The SYMD3 SET domain supported the trimethylation of H3K4, H4K5, and H4K20. These, in turn, managed transcriptional control by interacting with an RNA polymerase complex [3,31]. This type of enzymatic histone alteration was augmented by tumor-specific proteolysis of the SMYD3 N-terminal 34 residues to release the methylated protein substrate [31]. Since then, SMYD3 has been linked to the tumorigenic cascade for various types of cancer in which methyltransferase activity is dysregulated [30,31,113]. SMYD3 is overexpressed in hepatocellular carcinomas and colorectal carcinomas and contributes to the proliferation of breast carcinoma cells [3,30,31]. In the presence of overexpressed SMYD3, there have been 80 genes detected that exhibit altered gene expression. These include Nkx2.8, a homeobox transcription regulator gene dysregulated in hepatocellular cancers, oncogenes, and cell mediators [3,14,31]. 

The protein chaperone Hsp90 also plays a role in the epigenetic modifications by SMYD3 due to the catalytic stimulatory nature of the interaction [3,30,31]. The C-terminal domain is critical in generating SMYD3 enzymatic activity by serving as a binding motif for Hsp90 [30,113]. SMYD3 methylation of MAP3K2 induces oncogenic activation of the Ras signaling pathway, leading to tumorigenesis, as shown in Figure 8 [113]. Ras-driven carcinomas and their connection to enhanced expression of SMYD3 have been studied in pancreatic ductal adenocarcinomas in which aberrantly expressed SMYD3 plays a key role in tumorigenesis [113,114]. The presence of SMYD3 can be used for the diagnosis of cancer, as has been reported for thyroid and small cell lung cancer [115]. The inhibition of SMYD3 may prove to be desirable for the clinical management of various cancers [3,30,31,113,114]. 

#### 3.3.2. Therapeutic Applications, Potential Drug Targeting of SMYD3 and the Prognostic Value of SMYD3 

At this time, several major lead compounds have been reported to inhibit SMYD3 activity in vitro and in vivo. The structures of leads BCI-121, EPZ031686, EPZ030456, EPZ028862, GSK-49, and BAY-6035 are summarized in Figure 4 [116]. The expression and activity of SMYD3 was first reported in a preclinical model of colorectal cancer (CRC) in which SMYD3 was strongly upregulated throughout tumorigenesis at both the mRNA and protein level [42]. A virtual screening to identify new SMYD3 small molecule inhibitors was undertaken and reported in 2015. One of these compounds (BCI-121) significantly reduced SMYD3 activity both in vitro and in CRC cells, as suggested by the analysis of global H3K4me2/3 and H4K5me levels. These studies were extended to other cancers, including lung, pancreatic, prostate, and ovarian cancers where SMYD3 inhibitors were effective in all the tumor cell lines investigated [42]. 

One of the first orally bioavailable small molecules, reported in 2016, was the sulfonamide EPZ031686 (Figure 2 and Figure 4), a potent inhibitor of SMYD3 [46]. The X-ray structure of SMYD3 binding EPZ030456 and SAM was determined and the top two lead compounds were evaluated in detail for in vitro metabolic stability and permeability to determine their suitability for in vivo studies. With a mean scaled clearance of 34 mL/min/kg in mouse liver microsomal incubations, EPZ030456 was slightly less stable than EPZ031686 (24 mL/min/kg) under physiological conditions. EPZ030456 also had a lower apical-to-basolateral apparent permeability (P_app_ = 0.34 ± 0.22 × 10^−6^ cm/s) in Caco-2 cells than EPZ031686 (P_app_ = 0.64 ± 0.20 × 10^−6^ cm/s). Both compounds underwent active efflux in Caco-2 cells, with efflux ratios of 104 and 41, respectively. EPZ030456 and EPZ031686 had a free fraction of 0.32 ± 0.035 and 0.53 ± 0.12 in mouse plasma, respectively. Based on these compounds EPZ028862, an isoxazole sulfonamide, was developed that combined an improved potency of 1.8 nM with more favorable physicochemical properties and hence potentially suitable for evaluation in in vivo studies. Thus, overall, EPZ031686 had more favorable in vitro ADME profile than EPZ030456 and EPZ028862 has much potential [46]. 

Bay-6035 (Figure 4) [116], a benzodiazepine derivative, was identified using an SAR in which about 410,000 compounds were screened using a “Thermal Shift assay” [116]. The studies included the testing of racemic and chiral compounds using detailed SAR analysis to examine the interactions of the compounds from X-ray data. Variations in the amine side chain were considered, as were compound interactions with the lysine binding channel. Replacing the n-butyl chain with shorter or longer chains did not increase potency. However, because small cycloalkyl rings were one order of magnitude more effective, compounds containing the cyclopropyl ethyl and cyclobutyl ethyl side chains were examined in detail. The presence of three negatively charged amino acids (Glu 192, Asp 241, and Glu 294) close to the benzoic acid side chain in the crystal structure led to the investigation of compounds with a basic amine substituent and a wider range of urea derivatives with a range of protected diamines. Selecting the chiral S-enantiomers further improved the compound by an order of magnitude. Ultimately, the bridged azabicyclo [3.1.0] hexane-substituted urea compound (BAY-6035) was found to have a potency below 100 nM [116], making it the best compound identified in this screening assay.

The in silico screening identified lead compounds that were tested experimentally. However, only inhibitor 4 shown in Figure 9A was found to inhibit SMYD3. Figure 9B,C show the surface of SMYD3 (Figure 9B) and the interactions between inhibitor 4 and SMYD3 in the binding pocket of SMYD3 (Figure 4C). Figure 9C shows clearly inhibitor 4 is bound deeply inside the protein. Figure 9D shows the interactions between the inhibitor and hydrophobic and hydrophilic amino acids and Figure 9E shows the H-bonds between inhibitor and protein (in green) and supporting the high affinity of inhibitor 4. 

Reports continue to appear which identify new lead compounds [44,63,118,119]. One example is the identification of BAY-598 (Figure 5) [118] based on structural information from BAY-6035 (Figure 4). Most of the early leads were based on competitive inhibitors binding to the lysine substrate binding site. Because targeting the overproduction of SMYD3 is a desired outcome, some work has focused on the development of an irreversible inhibitor of the SMYD3 enzyme that covalently binds to the enzyme and directly reduces SMYD3 activity. There are reports of irreversible inhibitors that can be administered at lower concentrations and, as a result, are better tolerated (see inhibitor 29 in Figure 5) [44]. In another example, an in silico/in vitro hit-to-lead enzyme inhibition platform was used to identify a small molecular inhibitor for SMYD3 activity [43]. Searching libraries of small molecules for compounds with high binding affinity for SMYD3 led to the identification of an inhibitor, designated inhibitor 4 as shown in Figure 5, which disrupted uncontrolled oncogenic cell proliferation without affecting normal cell functioning in epithelial breast cancer, lung cancer, and colon cancer cell lines [43].

SMYD3 has been highly correlated, most notably with breast, hepatocellular, and colorectal carcinomas as well as with lung and pancreatic ductal adenocarcinomas [2,30,31,120]. The disease pathway for diethylnitrosamine (DEN)-induced hepatocellular and colorectal carcinomas involves damage to DNA and subsequent cell death, both of which are reduced in SMYD3-KO mice [120,121,122]. SMYD3 may also contribute to Ras-driven cancers as seen in elevated lysine methyltransferase activity in pancreatic malignancies [113,114,122]. In lung and pancreatic cancers, SMYD3 appeared to remain in the cell cytoplasm, where it functioned with the substrate MAP3K2 to enhance signaling through KRas to Erk1/2 [120]. Eliminating SMYD3 reduced the formation of pancreatic intraepithelial neoplasia and lung tumorigenesis, results which could attributed to a decrease in the phosphorylation of Erk1/2 in a mouse model [113]. The SMYD3 methylation of MAP3K2 on lysine 260 regulated MEK1/2 kinase signaling by hindering the binding of the PP2A serine/threonine phosphatase complex and subsequently decreased MAP kinase pathway inhibition, supporting the notion of a tumorigenic cascade [113,114]

### 3.4. SMYD4

#### 3.4.1. Structure and Function of SMYD4

Although the gene sequence of SMYD4 is significantly different from that of other members SMYD family, SMYD4 also functions as a lysine methyltransferase [81]. There is evidence for SMYD4 involvement in cardiac and muscle development in animal models, but the exact roles of SMYD4 remain poorly defined [81]. Studies in patients suffering from heart defects using target exome sequencing (TES) (c.1034G > A, p. G345D and c.1736G > A, p. R579Q) have identified potentially rare genetic variants of SMYD4 [81]. The analysis of 208 patients with congenital heart defects revealed the presence of two rare missense genetic variants of SMYD4 and suggested that genetic variants may contribute to congenital heart defects [81]. To examine this in more detail, the SMYD4(G345D) equivalent was induced in zebrafish using CRISPR/Cas9 to study its connection to heart development and more severe heart malformations in maturing embryos [81,82]. Another strategy which has been to evaluate the function of human SMYD4 and its *Drosophila* homologue and the sequences are presented in Figure 10 [77]. Human SMYD4m expressed in *Drosophila*, reduced transcription activity and was necessary for development [77,81]. The interaction of dSMYD4 with HDAC1 occurred at the N-terminus and was expressed in the mesoderm of the embryos [77]. A knockout of dSMYD4 led to death at the late pupal stage, suggesting that *Drosophila* dSMYD4 was necessary for development [77].

#### 3.4.2. Therapeutic Applications, Potential Drug Targeting and Prognostics of SMYD4

There is comparatively little information available on the role of SMYD4 in human health. A connection between SMYD4 and immunological responses has been shown for patients with rheumatoid arthritis [3,83]. Excess expression of SMYD4 appears in synovial fibroblasts stimulated with TNFα and has been observed in patients with this form of autoimmune disease [83]. There is growing evidence that SMYD4 has some role in tumor suppression [6,62,80]. A report on an oncogenic miRNA showed that miR-1307-3p functioned through targeting SET and MYND domains in SMYD4 [123,124]. Furthermore, this miRNA increased the proliferation of breast cancer cells and was overexpressed in gastric cancer cell lines. MiR-1307 is considered a risk factor for the development of metastatic types of cancers and its presence in serum can be used to diagnose breast cancer at the early stages of the disease [123].

### 3.5. SMYD5

#### Structure and Function of SMYD5

The structure and function of SMYD5 in mammalian tissues has not been studied extensively, although some information is available. SMYD5 has been linked to the immune system. It represses the cytokine transcription of genes encoding for the production of macrophages and primitive hematopoiesis and is involved in the differentiation of mouse embryonic stem cells (mESC) [3,4,86]. In addition, SMYD5 appears necessary for maintaining chromosomal integrity through the control of heterochromatin and the reduction of endogenous repetitions in DNA during differentiation [85]. For example, SMYD5 was recently reported to monoalkylate histone H3 on the lysine residue 36 or 37 (H3L36me1 or H3L37me1) in mammalian cells in vitro [4] and to trialkylate the H3 on the lysine residue 36 (H3L36me3) at promotors that regulate gene expression [5]. Several different enzymes catalyze methyl transfer [125]. H3K36me3 is generally formed by the catalysis of a different methyltransferase (SETD2) at the gene body regions while SMYD5 methylates lysines on the promotors, causing a change in conformation and thus affecting the regulation of the gene expression [5].

SMYD5 has been linked to the basal repression of a Toll-like receptor 4 (TLR4) response to a macrophage promoter that can detect pathogens within the organism [3,4] through the trimethylation of H4K20 (H4K20me3). H4K20me3 methylation occurs at repetitive LINE/LTR DNA sequences that recruit SMYD5 [4,87]. This mechanism involves the interaction of SMYD5 with nuclear receptor NCoR corepressor complexes to secure H4K20me3 marks on the promoters of target genes and the associated antagonization of TLR4-dependent gene activation [87]. With respect to heterochromatin regulation and endogenous retrovirus (ERV) silencing, SMYD5 impacts H4k20me3 through interaction with a chromatin repressor heterochromatin protein 1(HP1) and methyltransferase activity at H3K9, respectively (Figure 11) [87].

The role of SMYD5 in primitive hematopoiesis during embryogenesis has been studied using a zebrafish animal model [88]. Early in embryonic development, levels of SMYD5 are high and then decline steadily [88]. A study involving the loss of function of SMYD5 in zebrafish embryos demonstrated the average growth of gross morphological structures such as skeletal muscle and cardiac tissue, although there was a detected increase in pu.1, mpx, cymb, and l-plastin as well as in expressed primitive and definitive markers of hematopoiesis [88]. Studies of heterochromatin, marked by methylation of H4K20me3 regulated by SMYD5, suggest that there is a connection between this process and the pluripotency-associated self-renewal of embryonic stem cells [86]. Eliminating SMYD5 expression disrupted the self-renewal process, which involves the expression of OCT4 genetic targets affecting differentiation [86]. An SMYD5 knockout also caused a widespread decrease in H4K20me3, along with a reallocation of heterochromatin elements such as G9a, HP1α, and H3K9me3/2 and endogenous retroelement suppression [86]. SMYD5 has been recognized as a biomarker for hepatocellular carcinomas, where high expression is indicative of a poor patient prognosis [84].

## 4. Conclusions

Each member of the SMYD family of lysine methyltransferases contains a functional SET domain that is separated into two sections by the MYND (Myeloid-Nervy-DEAF1) domain, which contains a zinc-finger motif. Details of the properties and function of the various SMYD methyl transferases and the application of inhibitors targeting specific SMYDs are reviewed. Studies of the SMYD family of five different methyl transferases show the potential for the pharmaceutical development of SMYD inhibitors for the treatment and reversal of a number of diseases. This review outlines strategies for developing successful leads for inhibitors of SMYDs and describes some of these compounds. Several small molecule inhibitors targeting SMYDs in a variety of cancers and having low EC_50_ values have been identified. Many of these compounds have been used in cell and animal studies to provide information on their pharmacologic properties for use in vivo. Moreover, the inhibition or reversal of SMYD histone methylation suggests that treating epigenetic effects of SMYD protein overexpression will be a useful therapeutic approach. SMYD inhibitors also show promise in reversing drug resistance to known anticancer compounds, including doxorubicin and platinum-based drugs. Six epigenetic inhibitors have been developed, including inhibitors used to treat myelodysplastic syndrome (Vidaza, Dacogen), cutaneous T-cell lymphoma (Zoinza, Isrodax), and peripheral T-cell lymphoma (Beleodag, Epidaza). These drugs are currently in the clinic and suggest the promising use of inhibitors in the treatment pathologies associated with the SMYD enzymes. 

## Data Availability

The literature searches were conducted by A.P. and D.C.C. and was confirmed by D.A.R., M.A.B. and J.M.

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
