# Peer review of "Targeting Epigenetic Changes Mediated by Members of the SMYD Family of Lysine Methyltransferases"

_molecules, 2023, doi:10.3390/molecules28042000_

Round 1

Reviewer 1 Report

This review covers the medicinal and biological aspects of SMYD lysine methyltransferases.  The manuscript would be suitable for publication in Molecules once it addresses the following points:

i) The manuscript requires significant polishing. In the title, it should be Epigenetics (not Epigenesis). In the abstract, it should be epigenetic (not epigenitic). On page 2, it should be H3K36me1 and H3K36me3 (not H3L36me1 and H3L36me3). In the Conclusion, it should be zinc-finger motif (not zinc-figure motif). And many more. Also, there are many other places that have grammatical mistakes.

ii) The quality of figures is low. For example, text in Figure 1 is difficult to read. Please make higher resolution figures.

iii) When mentioning S-adenosyl-L-methionine on page 1, use both abbreviations (SAM, AdoMet), as both are used in the manuscript.

iv) Double check abbreviations of inhibitors. Labels in Scheme 2 and related text do not match.

v) There is no A) part in Figure 2.

vi) Table 1 lacks the caption. Replace ‘Drugs’ by ‘Inhibitors’.

vii) In Scheme 4, provide structures of inhibitors as mentioned in the caption.

Author Response

We thank the reviewer for the opportunity to address the comments by the reviewers. We have provided a copy (pdf) with Track Changes in a pdf so that all the changes made in the manuscript are indicated. This pdf is added as supplemental material. The version of the manuscript that is uploaded has all the changes accepted in the manuscript. Reviewer 1: The reviewer asks that we address specific concerns before the manuscript is suitable for publication. In particular the Reviewer requested “polishing” of the manscript and attention to editing. We have done this in the revised manuscript. ia) The Reviewer asks that we change the title of the manuscript to change Epigenesis to Epigenetics. The new title of the manuscript is “Targeting Epigenetic Changes Mediated by Members of the SMYD Family of Lysine Methyltransferases”. The use of epigenesis has also been changed in the Abstract as requested. Ib) On page 2, we have made the requested changes to H3K36me1 and H3K36me3. Zinc finger motif has been corrected in the Conclusion. We have removed many other grammatical mistakes as requested by the Reviewer. ii) The quality of all figures has been improved. Higher resolution images are provided as figures in a separate files. We kept screen shots of the revised figures manuscript to keep the size of the manuscript manageable. iii) We now use one abbreviation, SAM, for S-adenosyl-L-methionine throughout the manuscript. We have also changed the AdoMet label in Scheme 1 schematic. iv) Labels in Scheme 2 and the related text have been checked as requested and matched so only one abbreviation is used. v) The reference to Part A in Figure 2 has been removed. vi) A Title has been added to Table I and “Drugs” has been replaced by “inhibitors” in the revised manuscript most places. vii) The Reviewer requests the structures of inhibitors be mentioned in the caption of Scheme 4. This has been done as shown in the revised manuscript.

Reviewer 2 Report

This review article provides an overview on the SMYD family of methyltransferases and their disease association, with the focus on therapeutics development.

The SMYD family of methyltransferases play a vital role in various cellular processes, including gene expression, cell growth and proliferation. Several members of the SMYD family have been found to be overexpressed in various cancers, including SMYD2 and SMYD3. However, other diseases are also associated with SMYD methyltransferases that the authors didn't mention here. For example, SMYD1 has been shown to increase the levels of a protein called tau that is associated with the Alzheimer's disease (PMID: 34589886).

Comments:

·       Line 135, it was mentioned that leads were identified for SMYD1, SMYD2 and SMYD3. But further information about the leads are given for SMYD2 and 3 only. Would be good to include the lead structure for SMYD1 as well for comparison.

·       Line 184: the advantage of using in silico methods is to screen large library of compounds, often millions.

·       Line 194, “compounds interacting with the SAM binding pocket on SMYD would be avoided because compounds would not me specific.” I don't agree with this, as several selective SAM-competitive inhibitors were developed for other methyltransferases targeting the SAM binding pocket including PRMT5 and PRMT7.

·       Line 226: typo, “ bond” to “bound”

·       Line 225, x-ray structure of human SMYD1 is not solved, please indicate which organism.

·       Line 429 typo: "inhibits" written twice

·       Line 431: “that will mimic the binding properties of the p53 peptide may provide specificity to oncogenic SMYD2 function in p53-related cancers”.  This sentence sounds incorrect, consider rephrasing. In crystal structure of SMYD2 in complex P53 peptide (PDB: 3TG5), the p53 peptide occupies the same substrate binding pocket as other histone and non-histone peptides for methylation and targeting the substrate binding pocket in SMYD2 by small-molecules (even if it mimics the binding properties of p53 peptide) will inhibit the overall SMYD2 MTase activity and will not be p53-related cancer specific. 

·       Line 686: a figure presenting the co-crystal structure of protein with inhibitor would be beneficial for the reader to understand this paragraph.

Author Response

We thank the reviewer for the opportunity to address the comments by the reviewers. We have provided a copy (pdf) with Track Changes in a pdf so that all the changes made in the manuscript are indicated.  This pdf is added as supplemental material.  The version of the manuscript that is uploaded has all the changes accepted in the manuscript.

Reviewer 2:  The Reviewer suggests that we mention, as an example, the involvement of SMYD1 in increasing tau protein in Alziheimer’s Disease.  We have added this information to the revised manuscript and the appropriate citations are added.  Thank you – that is an important point and should be included in greater detail.

We have also addressed the specific comments made by Reviewer 2. 

  1. Line 135: The Reviewer asks that we include a lead structure for SMYD1 for comparison to lead structures for SMYD2 and SMYD3 in scheme 2. However, there are not yet reported inhibitors for SMYD1, SMYD4 and SMYD5 Ref. {6}. We have now added this in the caption but also a structure of a cancer drug (Doxorubicin) that does affect SMYD1 activity.

  1. Line 184: As requested, we include a statement indicating that the advantage of using in silico methods is to screen a large library of compounds in the revised manuscript.

  1. Line 194: The Reviewer indicates that several selective SAM competitive inhibitors have been developed for other methyltransferases targeting the SAM binding pocket including PRMT5 and PRMT7. We thank the reviewer for this comment and have revised the manuscript to reflect this information.

  1. Line 226 and Line 429: The typos have been corrected.

  1. Line 431: The Reviewer asks that we rephrase and correct this sentence. The new sentence is “that will mimic…” and appears on page X of the revised manuscript.  The reviewer provides guidance for this correction in the review.

  1. Line 688: The Reviewer suggests that presenting a crystal structure of the protein with inhibitor would be beneficial to the reader. We have now added two Figures (one for SMYD2 and one for SMYD3) in the revised manuscript binding peptides to show increased interactions between ligands and the SMYD protein.